# Use of Innovative Tools for the Detection of the Impact of Heat Stress on Reticulorumen Parameters and Cow Walking Activity Levels

**DOI:** 10.3390/ani13111852

**Published:** 2023-06-02

**Authors:** Ramūnas Antanaitis, Karina Džermeikaitė, Agnė Šimkutė, Akvilė Girdauskaitė, Ieva Ribelytė, Lina Anskienė

**Affiliations:** 1Large Animal Clinic, Veterinary Academy, Lithuanian University of Health Sciences, Tilžės Str. 18, LT-47181 Kaunas, Lithuania; karina.dzermeikaite@lsmu.lt (K.D.);; 2Department of Animal Breeding, Veterinary Academy, Lithuanian University of Health Sciences, Tilžės Str. 18, LT-47181 Kaunas, Lithuania; lina.anskiene@lsmuni.lt

**Keywords:** dairy cows, reticulorumen, heat stress

## Abstract

**Simple Summary:**

According to the literature, we hypothesize that the utilization of precision dairy farming technologies proves advantageous in the identification of heat stress indicators in dairy cattle. We found the influence of heat stress on reticulorumen parameters—increasing the risk of acidosis and the cows’ activity levels. Heat stress had a negative impact on reticulorumen pH, temperature, and rumination index. A higher THI (≥72) increases the risk of ruminal acidosis and cows’ physical activity levels. From a practical point of view, we can use innovative tools for the detection of heat stress and its impact on reticulorumen parameters and cow walking activity levels.

**Abstract:**

The aim of this study was to evaluate the influence of the temperature and humidity index on reticulorumen parameters such as temperature, pH, rumination index, and cow walking activity levels. Throughout the experiment, the following parameters were recorded: reticulorumen pH (pH), reticulorumen temperature (RR temp.), reticulorumen temperature without drinking cycles, ambient temperature, ambient relative humidity, cow walking activity levels, heat index, and temperature–humidity index (THI). These parameters were registered with particular smaXtec boluses. SmaXtec boluses were applied on 1 July 2022; 24 days were the adaptation period. Measurements started at 25 July 2022 and finished at 29 August 2022. The THI was divided into two classes: THI < 72 (comfort zone) and THI ≥ 72 (higher risk of thermal stress). Cows assigned to the 2nd THI class had lower average values for pH, temperature, and rumination index, but not walking activity levels. The mean differences ranged from 0.36 percent in temperature to 11.61 percent in walking activity levels (*p* < 0.01). An analysis of the THI revealed a significant positive linear relation with hours, where the THI had a tendency to increase on average by 0.2403. The reticuloruminal pH showed a negative linear relation with hours, where the reticuloruminal pH had a tendency to decrease on average by 0.0032, *p* < 0.001. Data analysis revealed a significant positive linear relationship between walking activity levels and hours, where walking activity levels had a tendency to increase on average by 0.0622 steps per hour, *p* < 0.001. The rumination index was not significantly related to hours (*p* < 0.005), although the rumination index had a tendency to increase by 0.4376 per hour, *p* > 0.05. The influence of HS on reticulorumen parameters increased the risk of acidosis and cows’ activity levels. HS had a negative impact on reticulorumen pH, temperature, and the rumination index. A higher THI (≥72) increased the risk of ruminal acidosis and decreased cows’ physical activity levels. From a practical point of view, we can use innovative tools for the detection of HS and its impact on reticulorumen parameters and cow walking activity levels.

## 1. Introduction

Heat stress (HS) is one of the most important variables influencing an animal’s immune system and production [1]. HS is the combination of environmental circumstances that cause an animal’s body temperature to rise as well as a physiological response. HS occurs when a cow’s endogenous heat production surpasses its capacity for heat dissipation [2]. The Intergovernmental Panel on Climate Change reported that the earth’s temperature has risen by 0.2 °C per decade, with the average temperature rising by 1.4 °C to 5.8 °C during the twenty-first century [3]. In dairy cattle, the thermoneutral zone—or temperature range at which a healthy adult animal may maintain a normal body temperature without spending energy above its usual basal metabolic rate—runs from 16 °C to 25 °C [4]. When an animal crosses its thermoneutral zone—which is a surface temperature of 22–25 °C in moderate weather and 26–37 °C in hot weather—increased heat gain as compared to heat loss from the body occurs, and the animal’s core body temperature begins to rise above the usual range, resulting in HS [5]. As a result, the Temperature Humidity Index (THI) is a potential indicator of HS. THI values greater than 72 are stressful for dairy cattle and are likely to negatively affect their welfare and output [1]. HS reduces an animal’s immunity and, as a result, has a direct impact on its health and wellbeing [1]. As HS causes a reduction in the amount of energy that dairy cows consume, these cows are unable to meet the demands that their bodies have for milk production and their overall health. Milk yields and quality are reduced as a result of this condition, and the animals are more susceptible to disease [6]. As a result, appropriate techniques for reducing HS in dairy cattle are essential and must be implemented on farms. Physical changes to the cow’s surroundings (such as providing shade and shelter, as well as cooling cows) and dietary interventions might help to mitigate some of the detrimental impacts of HS and may enhance dairy cattle health and output over the summer [1].

Precision dairy farming (PDF), or the use of technologies to monitor behavioural, physiological, or production markers for individual animal illness, oestrus, or comfort, is becoming increasingly popular [7]. Nowadays, PDF technology can monitor indicators such as laying time, rumination time, walking activity levels, temperature, and milk yield [8]. The behavioural and physiological differences discovered in this study shed light on the effects of breed, parity, milk yield (MY), and THI on cows [8]. An algorithm was recently created to remove drinking points from reticular temperature and link the reticulorumen fermentation temperature to the vaginal temperature using reticulorumen temperature (bolus sensor) [9]. This algorithm quantifies the relationship between vaginal and reticular temperature and allows for the reliable online continuous assessment of a cow’s body temperature using the ruminal bolus [9]. The boluses can measure both temperature and pH. Wireless boluses can transmit data every ten minutes, which can be saved in the cloud or on a PC. Measurements can be taken for up to a year depending on the battery life of the various bolus versions [10].

HS lowers rumen fermentation and milk production, making dairy cows particularly vulnerable to high environmental temperatures. When the cow’s normal cooling mechanisms—such as perspiration—are no longer effective, HS sets in. Strategies for preventing and treating heat exhaustion could be informed by studying how the body adjusts to different temperatures [11]. Lowered milk output, DMI, and walking activity levels are all the results of HS in cows [12]. Extensive monitoring is crucial for the early detection, investigation, and treatment of HS. There are numerous commercially available sensor systems for monitoring dairy cattle. Monitoring data acquired under HS indicate that rumination decreases as THI rises in cows [13]. High heat loads in the body of high-yielding cows during summer stress significantly alter feeding/drinking behaviour, digestibility, gut health, and nutrient transport across the blood–intestinal barrier. An increase in ambient temperature has a direct negative effect on the appetite centre of the hypothalamus and reduces feed intake. Heat-stressed cows ingest less feed and consequently ruminate less, resulting in less buffering substances (rumination is the main stimulus for saliva production) entering the rumen. In addition, due to the redistribution of blood flow to the periphery (in an attempt to improve heat dissipation) and the subsequent reduction in blood flow to the gastrointestinal tract, digestive end products (i.e., volatile fatty acids (VFA)) are absorbed less efficiently, resulting in an increased total VFA content in the rumen (and thus, a decreasing pH). Cows under thermally neutral conditions typically consume 12 to 15 meals per day, whereas the frequency of food intake with HS drops to three to five meals per day. This lower frequency is associated with larger meals and thus higher acid production after feeding. Chronic hyperthermia leading to severe or prolonged inappetence could also lead to subclinical and acute rumen acidosis [14].

According to the literature, we hypothesize that the utilization of precision dairy farming technologies could prove to be advantageous in the identification of heat stress indicators in dairy cattle. The aim of this study was to evaluate the influence of the temperature and humidity index on reticulorumen parameters such as temperature, pH, rumination index, and cow walking activity levels.

## 2. Materials and Methods

### 2.1. Animals, Housing, and Experimental Design

The provisions of the Lithuanian Law on Animal Welfare and Protection were followed in the conduct of this study. The study’s approval number is PK012858. This research was conducted on a Lithuanian dairy farm with 1500 cows; farm location—55.911381565736, 21.881321760608195. The cows were housed in a barn featuring open stalls and equipped with rubber mats. The milking process was carried out utilizing a DeLaval milking parlour. The animals were milked at 05:00 a.m. and 05:00 p.m. daily. The study was carried out over a period from 1 July 2022 to 1 September 2022. On the dairy farm, 21 clinically healthy dairy cows were enrolled in the trial at random.

The health status of these cows was checked every day by a local veterinarian. These cows were clinically healthy throughout the study period. Inclusion criteria—clinically healthy fresh Lithuanian Black and White breed dairy cows, average days in milk—30 (±5), second or more lactation, average milk production—32 (±5) kg/d., average body condition score—3.6 (±0.3; from 5-point scale), somatic cell count (SCC) levels in their milk were less than 180,000/mL (±0.55). Each cow attribute was extracted from the farm’s computer system and entered into a spreadsheet (lactation number, breed, latest calving date, and milk production; Delpro DeLaval Inc., Tumba, Sweden). The cows were kept in a loose arrangement year-round and fed total mixed rations (TMRs) at 6 a.m. and 6 p.m. daily and had ad libitum access to drinking water. High-producing, multiparous cows were fed a total mixed ration that largely comprised of 50% grain concentrate mash, alfalfa hay (13% protein), 10% grass silage, sugar beet pulp silage, 30% corn silage, 4% grass hay wheat straw, and compound feed. The ration’s chemical composition was as follows: 47.8% dry matter (DM); 29.02 (% of DM) neutral detergent fibre; 37.8 (% of DM) crude protein; acid detergent fibre non-fibre carbohydrates 17.5 (% of DM); and 1.8 (Mcal/kg) net lactation energy.

### 2.2. Measurements

Throughout the experiment, the following parameters were recorded: reticulorumen pH (pH), reticulorumen temperature (RR temp.), reticulorumen temperature without drinking cycles, ambient temperature, ambient relative humidity, cow walking activity levels, heat index, and temperature. These parameters were registered with particular SmaXtec boluses. The humidity index was calculated using the following formula: THI = 0.8*T + RH*(T − 14.4) + 46.4. The heat index was calculated using a heat stress calculator.

At the start of the study, all ten cows were given the SmaXtec boluses (SmaXtec animal care GmbH, Graz, Austria) orally. Boluses were used in accordance with the manufacturer’s instructions. The pH probes were calibrated at the beginning of the experiment using buffer solutions of pH 4 and pH 7. Daily statistics were recorded every 10 min. The SmaXtec messenger^®^ computer software collected and showed all the data. Qualified veterinarians implanted the rumen biosensor via the mouth, and data on rumen were recorded using software (SmaXtec, Austria). A SmaXtec climate sensor recorded the temperature and humidity on the farm.

#### Duration of Measurements

SmaXtec boluses were applied on 1 July 2022; 24 days was the adaptation period. Measurements were started at 25 July 2022 and finished at 29 August 2022. Reticulorumen parameters such as temperature, pH, rumination index, and cow activity levels were measured using SmaXtec boluses. The SmaXtec rumination metric indicates rumination activity levels over 24 h (24 h rolling sum), summed up in minutes. All these parameters are measured by a special SmaXtec metric system.

### 2.3. Data Analysis and Statistics

The records (*n* = 4446) of the tested cows (*n* = 21) in the summer season (5 weeks) were analysed using the IBM SPSS 25.0 (SPSS Inc., Chicago, IL, USA) program package. Using descriptive statistics, normal distributions were assessed using Kolmogorov–Smirnov test. The results were expressed as the mean ± standard error of the mean. The Pearson correlation was calculated to define the statistical relationships between the evaluated traits (rumination index, activity levels, reticulorumen temperature, pH, THI, temperature, and relative humidity). The one-way ANOVA test and general linear model–Repeated measures test were used for repeated measurements, including for time periods sing the same RumiWatch indicator, according to the days of the experiment. The THI was divided into 2 classes: THI < 72 (comfort zone) and THI ≥ 72 (higher risk of thermal stress). The LSD criterion was used to compare the differences in the mean between group values; a probability of less than 0.05 was considered significant (*p* < 0.05). Descriptive statistics of investigated indicators were carried out according to classes of THI [15].

## 3. Results

The data analysis of our research results revealed that cows assigned to the 2nd THI class had lower average values of pH, temperature, and rumination index, but not activity levels. The mean differences ranged from 0.36 percent in temperature to 11.61 in activity levels (*p* < 0.01; Table 1).

An analysis of the temperature humidity index (THI) revealed a significant positive linear relationship with hours, where THI had a tendency to increase on average by 0.2403 per hour, *p* < 0.001 (Figure 1a). A comparison of group means showed that the temperature humidity changed from 66.27 to 73.04 during the day; the range of changes was 9.27 percent, *p* < 0.001. Meanwhile, relative humidity showed a significant negative linear relationship with hours, where relative humidity had a tendency to decrease by 0.6658 per hour, *p* < 0.001 (Figure 1b); relative humidity changed from 58.47 to 78.60 during the day. The range of changes was 25.61 percent, *p* < 0.001. Data analysis revealed a significant positive linear relationship of temperature with hours, where temperature had a tendency to increase on average by 0.1994 °C per hour, *p* < 0.001(Figure 1c); the temperature changed from 19.66 to 25.35 during the day. The range of changes was 22.45 percent, *p* < 0.001.

Reticuloruminal pH showed a negative linear relationship with hours, where reticuloruminal pH had a tendency to decrease on average by 0.0032 per hour, *p* < 0.001 (Figure 1d); reticuloruminal pH changed from 6.21 to 6.56 during the day. The range of changes was 5.34 percent, *p* < 0.001. Reticuloruminal temperature showed a significant negative linear relationship with hours, where reticuloruminal temperature had a tendency to decrease by 0.0263 °C/h, *p* < 0.001 (Figure 1e); reticuloruminal temperature changed from 38.07 to 39.33 during the day. The range of changes was 3.20 percent, *p* < 0.001. Data analysis revealed a significant positive linear relationship of walking activity levels with hours, where walking activity levels had a tendency to increase on average by 0.0622 steps per hour, *p* < 0.001 (Figure 1f); walking activity levels changed from 4.34 to 9.09 during the day. The range of changes was 52.26 percent, *p* < 0.001. The rumination index was not related to hours (*p* < 0.005).

## 4. Discussion

We found that cows with a higher THI (≥72) had lower average values for reticulorumen pH, temperature, and the rumination index. Heat-stressed cattle often develop acidosis, in which a decrease in ruminal pH is observed—an actual problem during the summer months [15]. Moretti R. and other researchers have found that when THI increases, it decreases the frequency of rumination. Rumination is the main physiological function of cattle, so it is one of the most important indicators in the evaluation of animal welfare, and THI can be useful as an index of welfare evaluation [16]. With a high THI, it was seen that the animals spent less time ruminating and less time eating [17]. Given that forage digestion results in a significant amount of metabolic heat production, which raises body temperature, this could be seen as a behavioural adaptation to HS. Cows regulate their thermal balance and mitigate the impact of elevated external temperatures on heat transfer by reducing their feed intake—thereby preventing overheating. [17]. Dikmen and other researchers found that as the THI increases, the rectal body temperature of cattle also increases—with a THI of 78.2, the rectal temperature was 38.5 °C; with a THI of 81.5, the rectal temperature was 39.5 °C. When the THI was 83.6, the rectal temperature was 40.5 °C [18]. The conducted studies showed that when the THI increases, reticulorumen pH and temperature decrease. When the THI was <72, the reticulorumen temperature of the cattle was 39.43 °C. When the THI > 72, the reticulorumen temperature was 38.75 °C [19].

We found that reticuloruminal temperature showed a significant negative linear relation with hours, where reticuloruminal temperature had a tendency to decrease by 0.0263 °C, *p* < 0.001 (Figure 1e); reticuloruminal temperature changed from 38.07 to 39.33 during the day. The balance between heat production in the rumen and heat loss from the rumen affects rumen temperature [20]. There are seasonal shifts in cattle’s body temperature, which are reflective of the surrounding environment [21]. An increase in rectal temperature of 1 °C or less is sufficient to reduce performance in Holstein cows, so body temperature is a sensitive indicator of the cow’s physiological response to HS, as it is nearly constant under normal conditions [19]. Researchers have revealed that the body temperature of cattle may follow a circadian rhythm with a range of 0.2 to 0.9 °C, with the lowest temperature occurring in the morning and the highest occurring in the late afternoon. During the summer, both the average body temperature and the amplitude of the rhythm of body temperature are higher than during the spring and winter [21]. Kim, et al. found that the average and morning rectal temperatures of cows were higher (by 0.6 °C) during hot zone phases (30 °C) than the comfort temperature of 20 °C [22,23].

We found changes in reticulorumen pH. Reticuloruminal pH showed a negative linear relationship with hours, where reticuloruminal pH had a tendency to decrease on average by 0.0032/h, *p* < 0.001, and reticuloruminal pH changed from 6.21 to 6.56 during the day. The range of changes was 5.34%, *p* < 0.001. Real-time observations of the pH and temperature of the reticular contents of dairy cows have been suggested to be effective methods for assessing the risk of subclinical acidosis [24]. The examination of rumen contents is an extremely useful criterion for evaluating fermentation conditions, and the determination of reticular pH is the definitive test for diagnosing acidosis [25]. Cows that are more likely to be stressed by heat are more likely to have a lower reticulorumen pH [26]. HS can also reduce rumination, reticulo-rumen motility, and ruminal activity levels—all of which slow the fractional passage rate of digesta through the gastrointestinal system [27]. As a result of HS, the medial satiety centre of the hypothalamus is stimulated, which inhibits the lateral appetite centre—reducing nutritional intake and milk yield. By having a direct negative impact on the hypothalamic area that controls appetite, increased ambient temperatures decrease the amount of time spent ruminating and decrease appetite [28]. Cows undergoing HS consume less feed and ruminate less as a result, which decreases the number of buffering agents that reach the rumen (ruminating is the main stimulator of saliva production). Rumen acidosis is probably also caused by changes in cows’ eating habits. When temperatures are stable, cows normally eat 12 to 15 times daily; however, when under HS, they only eat three to five times daily [29]. Chronic hyperthermia can cause severe or persistent inappetence, and may also result in subclinical and acute rumen acidosis [30].

We found that THI had an impact on cows’ behaviour. Data analysis revealed a significant positive linear relationship of walking activity levels with hours, where walking activity levels increased on average by 0.0622 steps per hour, *p* < 0.001; walking activity levels changed from 4.34 to 9.09 during the day. The range of changes was 52.26 percent, *p* < 0.001. Cattle behaviour can change when the ambient temperature exceeds the animal’s comfort threshold, leading to HS—an early indicator of welfare, health, and productivity issues [31]. Heat stress can negatively impact reproductive function, leading to compromised fertility in females; this occurs through the inhibition of gonadotrophin-releasing hormone (GnRH) secretion in the hypothalamus, which is part of the hypothalamic–pituitary–gonadal (HPG) axis [21]. An increase in THI also increases the activity levels of the animal [32]. Animal welfare is negatively affected as a result of increases in activity levels, changes in feeding patterns, and decreased resting time when THI levels rise [17]. The study revealed that during days with high temperatures, bovine animals exhibited reduced levels of activity during the initial hours of the day, while displaying heightened levels of activity during the afternoon [33]. In a study carried out by Spanish scientists, it was proven that the resting time of animals experiencing HS was shorter—especially in the afternoon, where there was a noticeable decrease in resting time [17]. Heat-stressed cattle spend more time standing to increase heat loss through the skin [34]. Cattle activity level monitoring systems used in modern farms allow the assessment of cattle behaviour and, in case of deviations from normal behaviour, will send a warning to a computer program [35]. Sensors have recorded that cattle move their heads more during heat stress. It was found that cows take more steps per day in the summer months than in the cold season of the year [36]. Additional wellbeing issues are detected when changes in feeding, rumination, or standing/lying behaviours are significant enough to warrant a more thorough assessment [37]. HS has an impact on resting behaviour as well; cows rest for between 9 and 12 h a day on average, according to the literature. This leads to an average of 22 to 30 min every hour. This behaviour is a good measure of the welfare of cattle, as it is significantly affected when the animals are stressed or feel discomfort. When HS occurs, the amount of time spent resting each hour is decreased [17].

## 5. Conclusions

We found the influence of HS on reticulorumen parameters, increasing the risk of acidosis and cows’ behaviour. HS had a negative impact on reticulorumen pH, temperature, and the rumination index. A higher THI (≥72) increases the risk of ruminal acidosis and decreases cows’ physical activity levels. From a practical point of view, we can use innovative tools for the detection of HS and its impact on reticulorumen parameters and cow walking activity levels.

## Figures and Tables

**Figure 1 animals-13-01852-f001:**
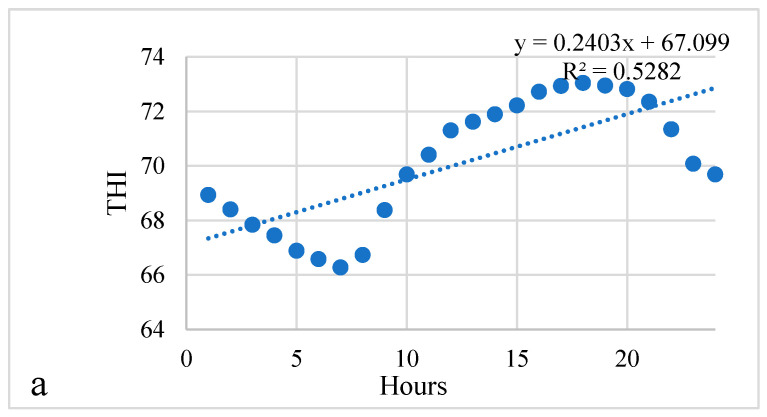
(**a**–**g**). Mean of the temperature humidity index, relative humidity, temperature, reticuloruminal pH, reticuloruminal temperature, walking activity levels, and rumination index during a 24 h period.

**Table 1 animals-13-01852-t001:** Data analysis according to THI classes.

Trait	THI Class	M	SE
Reticuloruminal pH, value	<72	6.34 ^A^	0.003
	≥72	6.32 ^B^	0.003
Reticuloruminal Temperature, °C	<72	38.85 ^a^	0.012
	≥72	38.70 ^b^	0.018
Activity levels, steps per hour	<72	5.87 ^A^	0.032
	≥72	6.77 ^B^	0.045
Rumination index, value average	<72	32,938.70 ^A^	27.229
	≥72	32,180.71 ^B^	41.365

THI class <72 *n* = 2917 records; THI class ≥72 *n* = 1529 records. Means with different superscripts within the same column represents significant differences between classes of THI: at *p* < 0.01 (A, B) or *p* < 0.05 (a, b) level.

## Data Availability

The data presented in this study are available within the article.

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
