# Peer review of "Use of Innovative Tools for the Detection of the Impact of Heat Stress on Reticulorumen Parameters and Cow Walking Activity Levels"

_animals, 2023, doi:10.3390/ani13111852_

Round 1
Reviewer 1 Report
The draft addresses an issue of relevance to dairy farming since heat stress affects both the productivity and well-being of dairy cows.
I have attached a series of observations that should be addressed in order to improve the document.
In lines 14 to 15 the authors wrote: “we hypothesize that there is an influence of the temperature and humidity index on reticulorumen parameters and cow walking activity registered with innovative tools”. However, the wording of the hypothesis seems incorrect to me, since in reality the research question focuses on whether PDF technologies are useful to identify stress indicators in dairy cattle, and not if that environmental variables like THI affects reticulorumen parameters and cow activity, which is well proven.
In line 17, the authors write “increasing the risk of acidosis and behavior of the cows”, but the expression increase for the behavior of the cows is not clear, because the authors do not explain how the increase could be recorded and subsequently, in line 19, state that they found a decrease in behavior,
On line 52, the authors clarify the acronym (HS) as an abbreviation for heat stress, which indicates that in the rest of the document this term will be referred to as HS, but it can be seen that throughout the document they write heat stress interchangeably or HS, which should be homogenized.
In lines 79 to 82, the authors write: "Nowadays, PDF technology can monitor indicators like as laying time, rumination time, walking activity, temperature, and milk yield. The behavioral and physiological differences discovered in this study shed light on the effects of breed, parity, milk yield (MY), and THI on cows [8]." In this regard, they are two citations from the same article that, when separated, cause confusion, since the reader can interpret that the second citation corresponds to the results of the article written by the authors, so it is suggested that it be rewritten in a single citation, avoiding confusion.
In lines 88 and 89 we can read: "Wireless boluses can transmit data every ten minutes. The data can be saved in the cloud 88 or on a PC". I suggest turning these two sentences into one... "Wireless boluses can transmit data every ten minutes, which can be saved in the cloud or on a PC."
At lines 105 to 107 the authors wrote: "According to the literature, we hypothesize that there is an influence of the temperature and humidity index on reticulorumen parameters and cow walking activity registered with innovative tools." The same suggestion is made as for the lines 14 and 15.
Within the registered parameters are indicated (line 140) among others, heat index and temperature - humidity index (THI), however they do not explain how these two parameters were calculated.
In lines 153 and 154 it is described that reticulorumen parameters such as temperature, pH, rumination index, and cow activity were measured using SmaXtec boluses, however, it is not explained how rumination index and cow activity were calculated.
In line 181 we can see the heading of Table 1, which does not cover the requirements of independently describing its content. On the other hand, it should be clarified what is the ruminal pH and the body or ruminal temperature, as well as in what units the traits were measured (°C or h, for example). Finally, at the bottom of the table it is stated that "Means with different 183 superscripts within the same column represents significant differences between classes of 184 THI: at p < 0.01 (A, B) or p < 0.05 (a, b) level.", however, lowercase literals are not observed in the table, only uppercase, so it is useless to clarify them.
Line 191 shows the bottom of figure 1, where three probability levels are indicated (*p<0.05,**p<0.01,***p<0.001, however, in the figure only ,***p<0.001, so it is unnecessary to record all levels; On the other hand, this figure must be edited, since the lines are overlapped with the text, which makes it difficult to observe it.
Figure 2 (line 193) is not mentioned before in the text, which is an error. In addition, the same recommendation of Figure 1 is made regarding the probability levels.
Table 2 (line 90) is also not mentioned before its presentation in the document, which must be done. In addition, it also lacks the information that allows it to be fully understood without reading the text of the document. The table also lacks in some column titles the values in which the variables are expressed: the percentage in Rel. humidity (which should not be abbreviated, it is better to write it down as RH), in RT temperature the °C and hours in activity. Finally, the literals should be listed in alphabetical order according to the value of the variable, in order to facilitate their understanding.
Line 205 indicates "Data analysis of our research revealed significant mean differences between weeks and 205 investigated traits", however, the statistical differences are only between the weeks of the study.
Lines 211 to 213 are used to discuss the differences between study weeks for the THI, addressed from a percentage point of view. However, the THI is measured in units and the authors considered a value equal to or greater than 72 as the one in which cattle become stressed and that value was obtained during weeks 3 and 4, that is, only in them is it considered that cattle may have suffered heat stress. The percentage differences in this indicator are not important.
Regarding figures 3, 4, 5 and 6, (reticulorumen pH, reticulorumen temperature, activity and, rumination index respectively), I consider that they do not correspond to the objective of the work, since in them the behavior of these indicators is observed according to the time of day, which should be the average value throughout the entire experimental period, but the authors do not associate it with ambient temperature or with the THI, so I consider they should be eliminated.
All the information in the first paragraph of the introduction, except the first sentence, corresponds to the introduction chapter, I don't see the need to include it in the discussion.
In line 276, we can read: "With a THI of 78.2, the rectal temperature of c is 276 38.5 °C...", and it is not clear what c means. On the other hand, all that sentence is not related to the study, since the maximum THI obtained in it was 72.36 and the quote refers to THI greater than 78.
How do you explain that, contrary to what the literature refers to, you report a higher reticulorumen temperature at THI less than 72 and less than >72? (lines 279-280).
In lines 294 and 295, the following is read: "Mean rectal and vaginal temperatures were greater in cows subjected to heat stress (40.0 and 40.0 °C, respectively)", noting that there is no range in the temperature data.
I do not find sense in the appointment of lines 297 to 300, since the effect of THI on the ruminal population was not measured.
The paragraph beginning at line 301 and ending at line 339 should not appear, as the study did not analyze daily changes in reticulorumen pH, reticulorumen temperature, activity and, rumination index respectively. The discussion should focus on the weekly periods, which is where the changes in the values of the variables indicated with those of relative humidity, ambient temperature and THI are shown.
Although it is indicated (line 340) that the cow activity changed from 3.94 340 to 9.28 steps per hour, this data does not appear in the results.
The information of line 354 is repeated from lines 343.
Author Response
Dear Reviewer,
Authors are very thankful for the comments, which help us to improve the manuscript. All changes proposed have been included in the manuscript and highlighted in yellow and track changes.
Best Regards,
Prof. Ramunas Antanaitis
|
Comments and Suggestions for Authors |
The responses and revisions provided. |
|
The draft addresses an issue of relevance to dairy farming since heat stress affects both the productivity and well-being of dairy cows. I have attached a series of observations that should be addressed in order to improve the document. |
We appreciate your comments. |
|
In lines 14 to 15 the authors wrote: “we hypothesize that there is an influence of the temperature and humidity index on reticulorumen parameters and cow walking activity registered with innovative tools”. However, the wording of the hypothesis seems incorrect to me, since in reality the research question focuses on whether PDF technologies are useful to identify stress indicators in dairy cattle, and not if that environmental variables like THI affects reticulorumen parameters and cow activity, which is well proven. |
In lines 14 and 15. The hypothesis has been modified to “According to the literature, we hypothesize that the utilization of precision dairy farming technologies proves to be advantageous in the identification of heat stress indicators in dairy cattle.” |
|
In line 17, the authors write “increasing the risk of acidosis and behavior of the cows”, but the expression increase for the behavior of the cows is not clear, because the authors do not explain how the increase could be recorded and subsequently, in line 19, state that they found a decrease in behavior, |
Line 17 we left the sentence - “We found the influence of heat stress on reticulorumen parameters, increasing the risk of acidosis and cows behavior”
Line 19 have been corrected to - “Higher THI (≥72) increases the risk of ruminal acidosis and cows physical activity.” |
|
On line 52, the authors clarify the acronym (HS) as an abbreviation for heat stress, which indicates that in the rest of the document this term will be referred to as HS, but it can be seen that throughout the document they write heat stress interchangeably or HS, which should be homogenized. |
The acronym "HS" referring to heat stress has been standardized throughout the entire manuscript. |
|
In lines 79 to 82, the authors write: "Nowadays, PDF technology can monitor indicators like as laying time, rumination time, walking activity, temperature, and milk yield. The behavioral and physiological differences discovered in this study shed light on the effects of breed, parity, milk yield (MY), and THI on cows [8]." In this regard, they are two citations from the same article that, when separated, cause confusion, since the reader can interpret that the second citation corresponds to the results of the article written by the authors, so it is suggested that it be rewritten in a single citation, avoiding confusion. |
Lines 79 and 82. This ciattion has been rewritten - “Nowadays, PDF technology can monitor indicators like as laying time, rumination time, walking activity, temperature, and milk yield. The behavioural and physiological differences discovered in this study shed light on the effects of breed, parity, milk yield (MY), and THI on cows [8]. |
|
In lines 88 and 89 we can read: "Wireless boluses can transmit data every ten minutes. The data can be saved in the cloud 88 or on a PC". I suggest turning these two sentences into one... "Wireless boluses can transmit data every ten minutes, which can be saved in the cloud or on a PC." |
Lines 88 and 89. The sentences was connected - "Wireless boluses can transmit data every ten minutes, which can be saved in the cloud or on a PC." |
|
At lines 105 to 107 the authors wrote: "According to the literature, we hypothesize that there is an influence of the temperature and humidity index on reticulorumen parameters and cow walking activity registered with innovative tools." The same suggestion is made as for the lines 14 and 15. |
Lines 105 and 107. The hypothesis has been modified to - “According to the literature, we hypothesize that the utilization of precision dairy farming technologies proves to be advantageous in the identification of heat stress indicators in dairy cattle.”
|
|
Within the registered parameters are indicated (line 140) among others, heat index and temperature - humidity index (THI), however they do not explain how these two parameters were calculated. |
Lines 185 and 186 - “Humidity index (THI) was calculated using the following formula THI =0.8*T + RH*(T-14.4) + 46.4.” Line 186 “Heat index was calculated using a heat stress calculator “ |
|
In lines 153 and 154 it is described that reticulorumen parameters such as temperature, pH, rumination index, and cow activity were measured using SmaXtec boluses, however, it is not explained how rumination index and cow activity were calculated. |
Lines 199 and 202 - “The smaXtec rumination metric indicates the rumination activity over 24 hours (24h rolling sum) summed up in minutes. All these parameters are measured by special smaXtec metric system” |
|
In line 181 we can see the heading of Table 1, which does not cover the requirements of independently describing its content. On the other hand, it should be clarified what is the ruminal pH and the body or ruminal temperature, as well as in what units the traits were measured (°C or h, for example). Finally, at the bottom of the table it is stated that "Means with different 183 superscripts within the same column represents significant differences between classes of 184 THI: at p < 0.01 (A, B) or p < 0.05 (a, b) level.", however, lowercase literals are not observed in the table, only uppercase, so it is useless to clarify them. |
Corrected |
|
Line 191 shows the bottom of figure 1, where three probability levels are indicated (*p<0.05,**p<0.01,***p<0.001, however, in the figure only ,***p<0.001, so it is unnecessary to record all levels; On the other hand, this figure must be edited, since the lines are overlapped with the text, which makes it difficult to observe it. |
Corrected. |
|
Figure 2 (line 193) is not mentioned before in the text, which is an error. In addition, the same recommendation of Figure 1 is made regarding the probability levels. |
Figure 1 and 2 corrected.
|
|
Table 2 (line 90) is also not mentioned before its presentation in the document, which must be done. In addition, it also lacks the information that allows it to be fully understood without reading the text of the document. The table also lacks in some column titles the values in which the variables are expressed: the percentage in Rel. humidity (which should not be abbreviated, it is better to write it down as RH), in RT temperature the °C and hours in activity. Finally, the literals should be listed in alphabetical order according to the value of the variable, in order to facilitate their understanding. |
Table 2 corrected |
|
Line 205 indicates "Data analysis of our research revealed significant mean differences between weeks and 205 investigated traits", however, the statistical differences are only between the weeks of the study. |
Line 205 was corrected to - “Data analysis of our research revealed significant mean differences betweenweeks.” and we left the statistical differences between the weeks. |
|
Lines 211 to 213 are used to discuss the differences between study weeks for the THI, addressed from a percentage point of view. However, the THI is measured in units and the authors considered a value equal to or greater than 72 as the one in which cattle become stressed and that value was obtained during weeks 3 and 4, that is, only in them is it considered that cattle may have suffered heat stress. The percentage differences in this indicator are not important. |
The sentece from Lines 211 to 213 have been deleated - “The significant mean difference in THI was detected between 5 th week (7.70 lower - compared to 5 thweek and 7.52 lower compared to 4 th week), (p<0.01).” |
|
Regarding figures 3, 4, 5 and 6, (reticulorumen pH, reticulorumen temperature, activity and, rumination index respectively), I consider that they do not correspond to the objective of the work, since in them the behavior of these indicators is observed according to the time of day, which should be the average value throughout the entire experimental period, but the authors do not associate it with ambient temperature or with the THI, so I consider they should be eliminated. |
Figures was deleted. |
|
All the information in the first paragraph of the introduction, except the first sentence, corresponds to the introduction chapter, I don't see the need to include it in the discussion. |
We deleted this paragraph |
|
In line 276, we can read: "With a THI of 78.2, the rectal temperature of c is 276 38.5 °C...", and it is not clear what c means. On the other hand, all that sentence is not related to the study, since the maximum THI obtained in it was 72.36 and the quote refers to THI greater than 78. |
Line 276. This sentece have been corrected to - “Dikmen S. and other researchers found that as the THI increases, the rectal body temperature of cattle also increases. With a THI of 78.2, the rectal temperature is 38.5 °C, with a THI of 81.5, the rectal temperature is 39.5 °C.” |
|
In lines 294 and 295, the following is read: "Mean rectal and vaginal temperatures were greater in cows subjected to heat stress (40.0 and 40.0 °C, respectively)", noting that there is no range in the temperature data. |
Line 294 and 295. The sentece was changed to - “Kim, K.H. et al. found that the average and morning rectal temperatures were higher (by 0.6°C) during the hot zone phases (30°C) than the comfort temperature of 20°C.” |
|
I do not find sense in the appointment of lines 297 to 300, since the effect of THI on the ruminal population was not measured. |
Line 297 and 300. The sentence “This involves a decrease in Treponema, Ruminococcus flavefaciens, Prevotella ruminicola, Flavonifractor, Fibrobacter succinogenes, and Flavonifractor [14].” was removed from the manuscript.
|
|
The paragraph beginning at line 301 and ending at line 339 should not appear, as the study did not analyze daily changes in reticulorumen pH, reticulorumen temperature, activity and, rumination index respectively. The discussion should focus on the weekly periods, which is where the changes in the values of the variables indicated with those of relative humidity, ambient temperature and THI are shown. |
We corrected this sentence with focus on HS. L407-430. |
|
Although it is indicated (line 340) that the cow activity changed from 3.94 340 to 9.28 steps per hour, this data does not appear in the results. |
Corrected |
|
The information of line 354 is repeated from lines 343. |
Information from line 343 was deleated - “Normally, cattle should rest for 9-12 hours [38].” |
Reviewer 2 Report
This manuscript addresses a problem with an increasing impact on global dairy herds, whether due to global warming or the increase of milk production in countries with hot climates. The design is based on the use of recent equipment that allows the monitoring of parameters of the cows, namely reticulorumen pH, temperature, and walking activity, and on the evaluation of their variation when subjected to different environmental conditions. The results reinforce the urgency of implementing more effective environmental control systems in preventing or reducing heat stress in dairy cows.
However, there are several aspects of this manuscript that could be improved.
Although the authors referred the geographical coordinates of the dairy farm, no reference is made to the environmental conditions of the farm, namely the type of housing (“loose arrangement”), bedding materials, whether the animals had access to an outdoor park, shading areas, environmental control systems, etc. Another shortcoming is the absence of a reference to the schedule and types of milking.
Considering the relatively quick response of the animals to environmental conditions, I don't think it makes any sense to indicate the variations of the parameters per week (lines 32-48 of the Abstract, Table 2, and lines 205-222 of the Results), even because the authors themselves refer that "Daily statistics were recorded every 10 minutes". To have an idea of the effects of heat stress over a longer period, it would perhaps be more interesting to know the impact on milk production, which is not addressed by the authors. For example, a heat stress episode can decrease milk yield for the next 2-3 days or even longer.
In Figures 1 and 2, I don't understand the need to correlate relative humidity and ambient temperature with the temperature-humidity index (THI), an index that is determined by these variables (single value representing the combined effects of air temperature and humidity).
I suggest the inclusion of a graph with the THI variation curve throughout the day together with the reticulorumen temperature (Figure 4) or other parameters to illustrate the effect of thermal stress.
The impact of heat stress on physical activity needs to be better clarified as there is some contradictory information in the manuscript.
There are several misspellings and grammatical errors throughout the text, as well as expressions of dubious meaning. Some examples:
- line 19, what do the authors mean by "decreases cows behavior"?
- line 117, please correct the sentence “From dairy farm 21 clinically healthy”
- lines 266-267 “Moretti R. and other researchers have also done research and found that when THI increases, so does the frequency of barking.”
- line 274, “cows lower their feed intake this helps them”
- line 276, “the rectal temperature of c is”
- line 282, “corelated”
- line 291, “0.9 degrees Celsius”
- lines 304, 307, 312 and 315, “retinal” instead of “reticular”
- line 345, “cattle are least active in”
Author Response
Dear Reviewer,
Authors are very thankful for the comments, which help us to improve the manuscript. All changes proposed have been included in the manuscript and highlighted in yellow and track changes.
Best Regards,
Prof. Ramunas Antanaitis
|
Comments and Suggestions for Authors |
The responses and revisions provided.
|
|
This manuscript addresses a problem with an increasing impact on global dairy herds, whether due to global warming or the increase of milk production in countries with hot climates. The design is based on the use of recent equipment that allows the monitoring of parameters of the cows, namely reticulorumen pH, temperature, and walking activity, and on the evaluation of their variation when subjected to different environmental conditions. The results reinforce the urgency of implementing more effective environmental control systems in preventing or reducing heat stress in dairy cows. However, there are several aspects of this manuscript that could be improved. |
Thank You for positive comments. |
|
Although the authors referred the geographical coordinates of the dairy farm, no reference is made to the environmental conditions of the farm, namely the type of housing (“loose arrangement”), bedding materials, whether the animals had access to an outdoor park, shading areas, environmental control systems, etc. Another shortcoming is the absence of a reference to the schedule and types of milking. |
The manuscript has been corrected and added information regarding the environmental conditions of the farm, specifically pertaining to the type of housing utilized, which is described as a "loose arrangement," as well as the materials used for bedding. Additionally, the schedule and methods of milking were included.
Added information: “The cows were housed in a barn featuring open stalls and equipped with rubber mats. The milking process was carried out utilizing a DeLaval milking parlor. The animals were milked at 05:00 a.m. and 05:00 p.m. daily. The cows were protected from the harmful effects of the sun's rays, precipitation, wind, and mud since they were kept in a barn that had a full roof and fans that turned on automatically when the temperature reached 25 °C. Animals hadn't access to an outdoor park.”
|
|
Considering the relatively quick response of the animals to environmental conditions, I don't think it makes any sense to indicate the variations of the parameters per week (lines 32-48 of the Abstract, Table 2, and lines 205-222 of the Results), even because the authors themselves refer that "Daily statistics were recorded every 10 minutes". To have an idea of the effects of heat stress over a longer period, it would perhaps be more interesting to know the impact on milk production, which is not addressed by the authors. For example, a heat stress episode can decrease milk yield for the next 2-3 days or even longer. |
Thanks for your observations and insights. The aim of our study was to determine the influence of heat stress, namely for shorter time intervals. The amount of milk during our study was not recorded and analysed.
|
|
In Figures 1 and 2, I don't understand the need to correlate relative humidity and ambient temperature with the temperature-humidity index (THI), an index that is determined by these variables (single value representing the combined effects of air temperature and humidity). |
Figure 1 and 2 was corrected |
|
I suggest the inclusion of a graph with the THI variation curve throughout the day together with the reticulorumen temperature (Figure 4) or other parameters to illustrate the effect of thermal stress. |
Figure 4 was dele |
|
The impact of heat stress on physical activity needs to be better clarified as there is some contradictory information in the manuscript. |
Lines 474 and 476 added information: “Animal welfare is negatively affected as a result of an increase in activity, changes in feeding patterns, and a decrease in rumination and resting when THI levels rise. “
Lines 486 and 488 added information: “ Sensors have recorded that cattle move their heads more during heat stress. It was found that cows take more steps per day in the summer months than in the cold season of the year.”
Lines 473 and 476 added information: “ Heat stress can negatively impact reproductive function, leading to compromised fertility in females. This occurs through the inhibition of gonadotrophin releasing hormone (GnRH) secretion in the hypothalamus, which is part of the hypothalamic-pituitary-gonadal (HPG) axis. “
|
|
Comments on the Quality of English Language |
|
|
There are several misspellings and grammatical errors throughout the text, as well as expressions of dubious meaning. Some examples: |
We have corrected the following lines in the manuscript: - Line 19 - “Heat stress had a negative impact on reticulorumen pH, temperature, and rumination index. Higher THI (≥72) increases the risk of ruminal acidosis and decreases cows physical activity.” - Line 117 - “On a dairy farm, 21 clinically healthy dairy cows were enrolled in the trial at random.” - Lines 266-267 - “Moretti R. and other researchers have also done research and found that when THI increases, decreases the frequency of rumination.” - Line 274 - “With a THI of 78.2, the rectal temperature is 38.5 °C” - Line 282 - “correlated” - Line 291 - “with a range of 0.2 to 0.9 °C” - Lines 304, 307, 312 and 315 - all words changes to “reticular” - Line 345 - “The study revealed that during days with high temperatures, bovine animals exhibit reduced levels of activity during the initial hours of the day, while displaying heightened levels of activity during the afternoon” |
Round 2
Reviewer 1 Report
Thank you for taking into account the comments and suggestions.Author Response
Dear Reviewer,
The authors are very thankful for the positive opinion.
Best Regards,
Prof. Ramunas Antanaitis
Reviewer 2 Report
In lines 34-50, I still don't understand the relevance of the reference to the change of the variables per week since the studied effects, such as physical activity and rumination index, are affected in a short period of time, hours or even minutes.
In lines 100-104, the authors should better clarify the link between heat stress and ruminal acidosis since the associated drop in dry matter intake can contribute to the increase in ruminal pH, which goes in the opposite direction to what the authors state, which may confuse the reader.
In lines 123-126, the content and the way it is written conveys the idea that cows should be always stabled, which goes against all current recommendations that advocate that cows should be outdoors as much as possible, except in extreme weather conditions. I don't think the authors have that conviction, but one gets that impression after reading the sentence.
In line 204, I don't understand the relevance for the journal's scope of stating that "Relitive humidity was highly negatively corelated with temperature". By the way, replace "relitive" with “relative" and “corelated” with “correlated”.
In Figure 2, I don't understand the logic of correlating THI with temperature and relative humidity, an index that is calculated with these two variables.
I still don't think it is relevant to indicate the means, standard errors, etc., of the investigated indicators according to weeks, as shown in Table 2 and in the paragraph between lines 235 and 254.
In lines 301 and 302, it is mentioned that the authors "found that THI was highly positively correlated with reticulorumen temperature" but, according to the previously written (e.g., Figure 2), the opposite was verified.
Lines 337-340, repeated information? If not, please clarify.
Moderate editing of English language required.
Author Response
Dear Reviewer,
Authors are very thankful for the comments, which help us to improve the manuscript. All changes proposed have been included in the manuscript and highlighted in yellow and track changes.
Best Regards,
Prof. Ramunas Antanaitis
|
Question |
Answers |
||||||||||||||||||||||||||||||||||||||||||||||||
|
In lines 34-50, I still don't understand the relevance of the reference to the change of the variables per week since the studied effects, such as physical activity and rumination index, are affected in a short period of time, hours or even minutes.
|
Corrected. Analysis was made according to hours (24 hour / day). Test for a linear trend in the investigated traits means, based on the values for the factor levels: hours was applied. Means were calculated and ANOVA Trend Analyses Polynomial was conducted (results in a table below); after that it was decided to show the relation of time change with investigated trait, using linear regression analysis.
P< 0.001 considered significant; the line can reliably describe the change in means.
|
||||||||||||||||||||||||||||||||||||||||||||||||
|
In lines 100-104, the authors should better clarify the link between heat stress and ruminal acidosis since the associated drop in dry matter intake can contribute to the increase in ruminal pH, which goes in the opposite direction to what the authors state, which may confuse the reader. |
We added information – “High heat loads in the body of high-yielding cows during summer stress significantly alter feeding/drinking behavior, digestibility, gut health, and nutrient transport across the blood-intestinal barrier. An increase in ambient temperature has a direct negative effect on the appetite center of the hypothalamus and reduces feed intake. Heat-stressed cows ingest less feed and consequently ruminate less, resulting in less buffering substances (rumination is the main stimulus for saliva production) entering the rumen. In addition, due to the redistribution of blood flow to the periphery (in an attempt to improve heat dissipation) and the subsequent reduction in blood flow to the gastrointestinal tract, digestive end products (i.e., volatile fatty acids (VFA)) are absorbed less efficiently, resulting in increased total VFA content in the rumen (and thus a decreasing pH). Cows under thermally neutral conditions typically consume 12 to 15 meals per day, whereas the frequency of food intake at HS drops to 3 to 5 meals per day. The lower frequency is associated with larger meals and thus higher acid production after feeding. Chronic hyperthermia leading to severe or prolonged inappetence could also lead to subclinical and acute rumen acidosis [14]”
|
||||||||||||||||||||||||||||||||||||||||||||||||
|
In lines 123-126, the content and the way it is written conveys the idea that cows should be always stabled, which goes against all current recommendations that advocate that cows should be outdoors as much as possible, except in extreme weather conditions. I don't think the authors have that conviction, but one gets that impression after reading the sentence. |
We deleted this sentence – “The cows were protected from the harmful effects of the sun's rays, precipitation, wind, and mud since they were kept in a barn that had a full roof and fans that turned on automatically when the temperature reached 25 °C. Animals hadn't access to an outdoor park” |
||||||||||||||||||||||||||||||||||||||||||||||||
|
In line 204, I don't understand the relevance for the journal's scope of stating that "Relitive humidity was highly negatively corelated with temperature". By the way, replace "relitive" with “relative" and “corelated” with “correlated”. |
We deleted this information |
||||||||||||||||||||||||||||||||||||||||||||||||
|
In Figure 2, I don't understand the logic of correlating THI with temperature and relative humidity, an index that is calculated with these two variables. |
We deleted this figure and information. |
||||||||||||||||||||||||||||||||||||||||||||||||
|
I still don't think it is relevant to indicate the means, standard errors, etc., of the investigated indicators according to weeks, as shown in Table 2 and in the paragraph between lines 235 and 254.
|
We added additional calculations based on hours. |
||||||||||||||||||||||||||||||||||||||||||||||||
|
In lines 301 and 302, it is mentioned that the authors "found that THI was highly positively correlated with reticulorumen temperature" but, according to the previously written (e.g., Figure 2), the opposite was verified. |
Sorry, it was not changed. It was detected low negative relation of THI with reticulorumen temperature ( r= - 0.076, p<0.001). |
||||||||||||||||||||||||||||||||||||||||||||||||
|
Lines 337-340, repeated information? If not, please clarify.
|
We deleted repeated information |
||||||||||||||||||||||||||||||||||||||||||||||||
|
Moderate editing of English language required. |
Corrected in whole manuscript |